# Design, Implementation and Evaluation of an Innovative Pilot Intervention to Improve the Family Quality of Life of Children with Specific Learning Disabilities—A Randomized Controlled Trial

**DOI:** 10.3390/ijerph20247192

**Published:** 2023-12-18

**Authors:** Nektaria Pedioti, Stavroula Lioliou, Katerina Koutra, Stavros Parlalis, Maria Papadakaki

**Affiliations:** 1Department of Social Work, School of Health Sciences, Hellenic Mediterranean University, 71004 Heraklion, Greecempapadakaki@hmu.gr (M.P.); 2Department of Psychology, Faculty of Social Sciences, Gallos Campus, University of Crete, 74100 Rethymno, Greece; 3Department of Psychology and Social Sciences, School of Education and Social Sciences, Frederick University, Nicosia 1036, Cyprus

**Keywords:** specific learning disability, children, experimental design, quality of life

## Abstract

**Background**: The high prevalence of learning disabilities among children confirm that learning disabilities are surprisingly common. In the absence of routine screening, many children still go undetected with a huge individual and family burden, while at the same time, the findings of existing interventions are conflicting. This study reports on the design, implementation and evaluation of an innovative pilot intervention aiming at improving the quality of life of the family of children with specific learning disabilities. **Method**: For the purposes of this study, we ran a randomized controlled trial employing an experimental research design with two groups (intervention and control). The study population comprised parents of children with specific learning disabilities. Out of the 71 individuals that were eligible for randomization, 42 were allocated to the intervention, and 29 to the control group. A brief parenting intervention model was employed, aiming at improving parenting skills through a stepwise process. The intervention included four skill building sessions conducted over a period of 6 weeks. “Parenting style” (including three dimensions: “Authoritative”, “Authoritarian” and “Permissive”) and “Family Quality of life” (including five dimensions: “Family Interaction”, “Parenting”, “Emotional Well-being”, “Physical/Material Wellbeing” and “Disability-Related Support”) were employed as the outcome measures of this study. Two validated questionnaires were used to measure the study outcomes: “the Parenting Style Questionnaire” and the “Family Quality of Life Scale (FQOL) Questionnaire”. The questionnaires were applied at the pre- and post-intervention level. **Findings**: An analysis showed that except for the “permissive parenting style”, the intervention and control group had statistically significant differences in all the “Parenting style” and the “Quality of life” dimensions at the post-intervention level. In the intervention group, none of the study dimensions improved at a statistically significant level at the post-intervention level compared to pre-intervention level. According to the cluster analysis, which re-examined successful vs. unsuccessful cases, the intervention was found to have an effect on the average values of all the “quality of life” and “parenting style” dimensions, except for the “Authoritarian Parenting Style”. **Conclusions**: The study offers evidence on the dimensions of parenting and quality of life mostly affected by a brief intervention as well as on the feasibility, practicality and acceptance of such interventions in local communities.

## 1. Introduction

A specific learning disability (Sp.L.D.) refers to a condition, which results in difficulties in one or more essential psychological processes related to language comprehension and usage, both spoken and written, involving multiple challenges in tasks such as listening, thinking, speaking, reading, writing, spelling and performing mathematical calculations [1]. Sp.L.D.s do not encompass learning difficulties that primarily arise from visual, hearing or motor disabilities, intellectual disability, emotional disturbance or adverse environmental, cultural or economic circumstances [2]. Over 4 million children in the US have at least one learning disability [3]. Nearly 1.69% of children live with one or more learning disabilities, while 20% have learning and attention problems [4]. Currently, determining the exact prevalence of Sp.L.D.s. in the EU-28 is a challenging task due to two key factors: the co-occurrence of Sp.L.D.s. with other conditions and the lack of comprehensive available statistics across the EU-28 region. In 2012, there were approximately 41 million individuals within the primary and secondary education systems of the EU-28 who had an Sp.L.D. yet remained unidentified and without adequate support [5]. Today, the percentage of Sp.L.D.s in Greece varies between 5 and 12%, with this variation explained by the varied definitions used in surveys [6].

A supportive environment can positively regulate the learning behavior of children with an Sp.L.D., and parents have an important role to play [7]. They are the primary caregivers for children with an Sp.L.D., supporting and promoting their academic success and strengthening their self-confidence and self-determination. Nevertheless, it is often the case that they lack sufficient knowledge and awareness regarding Sp.L.D.s to timely recognize and efficiently respond to these “hidden” conditions. At times, they may attribute early signs to normal developmental processes and delay intervention [7]. Moreover, on certain occasions, they may find it challenging to acknowledge their children’s difficulties, even when teachers or significant others attempt to bring them to their attention [8]. It has been suggested that, in some cases, parents may tend to hold a more skeptical stance towards the issue, which can encompass elements of denial and reluctance to accept it. Such reactions may result from the difficulties parents encounter when trying to establish reasonable expectations for their child. These challenges have been shown to have an adverse impact on parents, potentially subjecting them to heightened levels of stress, frustration, and dissatisfaction and sometimes resulting in a tendency towards overprotectiveness [8]. This, in turn, has been shown to impact their marital satisfaction, mental health status, psychological well-being and overall quality of life [9,10,11]. 

### 1.1. Literature Review

Various interventions have been developed aiming at improving parenting styles and parent–child interaction. For example, a positive shift in the engagement level of parents with children with Sp.L.D.s and behavioral, emotional and social challenges was reported by Lendrum et al. [12], who delivered a comprehensive overview of the key outcomes from the Achievement for All (AfA) program. This transformation was attributed to the constructive dialogues between schoolteachers and parents, which replaced the prior negative feedback loop focused on behavioral problems and incidents. Another effective intervention was based on Family Check-Up (FCU), which is an evidence-based intervention program designed to promote positive family dynamics and improve child behavior and well-being. It is rooted in the principles of family systems theory and focuses on enhancing parenting skills, communication, and overall family functioning. While the FCU can be used to address existing behavioral issues, it also has a preventive focus. By strengthening family relationships and parenting skills, it aims to reduce the risk of future problems [13,14]. In Greece, interventions are few and limited in scope. Besides few recent efforts in the educational setting aiming to support the students’ learning through strengthening the cooperation between the teachers and the parents, there are no other initiatives targeting parenting style through engaging both parents in a structured intervention program [15]. Such interventions are highly warranted in the Greek setting as they are thought to have a valuable impact on promoting the development of social and emotional skills during the early stages of a child’s growth, while at the same time they hold great potential in positively influencing children’s social interactions, enhancing cognitive and behavioral abilities and improving their academic accomplishments [16].

### 1.2. Research Goals and Questions

This study aimed to evaluate the effectiveness of a new intervention program in improving both parenting styles and the overall family quality of life. This study aimed to answer the following research questions: does the intervention improve parenting style? Does it improve family quality of life?

## 2. Materials and Methods

### 2.1. Study Design

This is a randomized controlled trial with two-groups (intervention and control) and a pre- and post-evaluation design. The current paper reports on the data collected at baseline and immediately after the intervention, and an annual follow up has been planned.

### 2.2. Participants’ Recruitment

Study participants were recruited from a population that used a community-based university-led mobile unit in charge of evaluating children for learning disabilities across the Crete Region of Greece. This unit is unique in Greece and is accessible through the Social Services Departments of the local municipalities. Individuals interested in having their children evaluated may apply for the service on a voluntary basis without charge. Those eligible to participate in the current study were those that fulfilled the following criteria: (a) parents of children in the 3rd to 6th grade, (b) children with an Sp.L.D. diagnosis, (c) residents of the Crete region, (d) competency in the Greek language and (e) availability and willingness to participate in the study. In total, 71 individuals fulfilled the inclusion criteria out of the 158 evaluated by the mobile unit. Two independent researchers performed the eligibility check. The participants were randomly allocated to the intervention and control group. Randomization was performed via a computer-generated list of random numbers. The list was then re-sorted to its original order. Randomization was conducted by two researchers, who were neither involved in the eligibility check nor performed data collection. Out of the 71 individuals that were eligible for randomization, 42 were allocated to the intervention group (to receive the 6 week parenting program), and 29 to the control group (to receive no intervention/be added to wait list for the intervention). Upon allocation to the groups, three individuals dropped from the intervention group and 20 from the control group, primarily due to limited availability, health problems and change of residence (see Figure 1). Adherence to the study was calculated to be 67.6% (individuals who completed the intervention against those lost in follow up). None of the sessions were missed by our intervention group. Two individual researchers were blinded to the intervention allocation and reviewed the results, and two independent analysts in charge of data analysis were blinded to the questionnaire administration and the data collection process. 

Prior to participation, individuals were requested written consent be provided upon being offered information on the study aim and procedures and upon receiving guarantees about confidentiality, protection of their personal data and the right to withdraw at any time during the intervention.

### 2.3. Outcome Measures 

To assess the quality of life and the parenting style of the participants, two self-administered validated questionnaires were used, which were translated into the Greek language and free of charge with secured permission by the developers: The Parenting Style Questionnaire [17]. This questionnaire has good psychometric properties and has been translated and used in several studies such as Matos et al. [18], Önder & Gülay [19] and Tagliabue et al. [20]. It contains 32 items such as “I do praise my child”, “I punish my child by taking away privileges”, “I have outbursts of anger”, “I have difficulty disciplining him”, etc. Participants were invited to indicate how frequently they perform each parenting behavior. Responses ranged between 1 (never) and 5 (always). Out of the 32 items, 13 items form the “Authoritarian parenting style”, 15 items form the “Authoritative parenting style” and 4 items form the “Permissive Parenting style”. A composite score was calculated for each parenting style (ranging from 13 to 65 for “Authoritarian style”, 15 to 75 for “Authoritative style” and 4 to 20 for “permissive style”), with higher scores indicating a higher alignment of the participants to the respective parenting style.The Family Quality of Life Scale (FQOL) questionnaire [21]. This questionnaire has been successfully used in previous studies such as Poston et al. [22] and Park et al. [23]. It contains 25 items such as “in your family do you help children learn to be independent”, “in your family do you teach children how to get along with others”, “in your family do you support each other in achieving goals”, etc. Participants were invited to indicate how satisfied they were with each statement reflecting their family life. Responses ranged between 1 (very dissatisfied) and 5 (very satisfied). Out of the 25 items, 6 items form the “Family Interaction”, 6 items form the “Parenting”, 4 items form the “Emotional Well-being”, 5 items form the “Physical/Material Well-being” and 4 items form the “Disability-related support”. A composite score was calculated for each parenting style (ranging from 6 to 30 for “Family Interaction”, 6 to 30 for “Parenting”, 4 to 20 for “Emotional Well-being”, 5 to 25 for “Physical/Material Well-being” and 4 to 20 for “Disability-related Support”), with higher scores indicating higher satisfaction with their family life.

Besides the above-mentioned study questionnaires, a semi-structured interview schedule was designed based on past research to explore parents’ attitude and knowledge [24,25]. Moreover, a special intake form was developed and used during the initial assessment to capture information on the participants’ individual, family and social circumstances.

Pre- and post-intervention questionnaires, assessing the outcome measures of interest (parenting style and quality of life) were collected through online forms that were delivered by members of the research team.

### 2.4. Content of the Intervention

The current study used, upon permission, an adapted version of Cipani’s [26] model. This consists of a 6 week program in which trained mental health professionals work with parents to enhance their basic skills in addressing children’s negative behavior. The program involves individual and group sessions with parents and is particularly suitable for children at the preschool and elementary-school level. For the purposes of our study, we used only individual sessions with parents. The intervention was delivered by two mental health professionals with a bachelor’s degree in social work or psychology, postgraduate studies in special education and special training in family counseling. In our study, the program included four sessions with participants in the intervention group, a two-phase design and a total duration of six weeks. The first phase included one clinical session with a 2 h duration for acquaintance with the parents as well as an exploration of their attitudes and knowledge about Sp.L.D.s. The second phase included three sessions (2 h duration each) focusing on the parents’ relationship with their child and reinforcing positive parenting practices through a stepwise learning process and a number of home-based assignments assigned on a weekly basis. The first session was face-to-face (on university premises), while the 2nd, 3rd and 4th sessions employed a video call for communications between the mental health professional and the parents. These assignments involved a training element with a focus on specific communication techniques (i.e., physical proximity and face-to-face orientation), positive disciplinarian techniques (i.e., “praise”, time limits, “sit and decide” as well as the “non-compliance jar”) and other parent–child interaction facilitators. Parents were initially assessed in terms of their beliefs regarding their children’s level of compliance using a 6-item scale with responses ranging from “Always/All the time” to “Never/Not at all” (e.g., “I find myself repeating the request multiple times” “I have to scream at him/her to get compliance”). They were re-assessed at the end of the 6 week program to identify differences in compliance levels after the intervention. During the intervention, parents were trained on self-observation and were asked to record their children’s compliance levels when different commands/requests were given, during the weekly assignments, using a structured diary with 5 items per command/request (i.e., “Time of the day the command was given”, “Type of command”, “Technique used”, “Quality of interaction/communication” and “Level of compliance”). At the end of the intervention, parents were asked to self-evaluate their performance using a standard evaluation sheet, which summarized the successes and failures and offered a total achievement level, which was reported quantitatively (“% of compliance”) and qualitatively (“No improvement”, “Modest improvement”, “Sufficient improvement” and “Dramatic improvement”). The self-assessment result was then discussed in consultation with a mental health professional during a follow up session, in which errors and corrective actions were identified and processed.

### 2.5. Statistical Analysis

A data analysis was conducted using Statistical Package for Social Sciences (SPSS) version 25.0 software. The significance level of this study was set at α = 0.05. We employed a combination of parametric and non-parametric tests to assess the mean differences between two groups, selecting the appropriate test based on the normality distribution of the data. For statistical inference, the non-parametric equivalents of independent *t*-tests and paired *t*-tests were used in case the parametric assumptions were not met.

The dimensions of the “quality of life” questionnaire (“Family Interaction”, “Parenting”, “Emotional Wellbeing”, “Physical/Material Wellbeing” and “Disability-Related Support”) and “parenting style” questionnaire (“Authoritative Parenting Style”, “Authoritarian Parenting Style” and “Permissive Parenting Style”) were calculated, with the control and exclusion of cases from the sample that did not meet the criteria for correct completion.

Specifically, the analysis included the following steps: Comparisons of dimensions of “quality of life” and dimensions of “parenting style” scores were implemented, before and after the intervention, mainly for the intervention group, using paired *t*-tests;Comparisons of “quality of life” and “parenting style” scores were implemented before the intervention between the intervention and the control group. We used an independent samples *t*-test. However, there were many withdrawals from the second test from control group, and the results after the intervention were unstable for statistical conclusions.

First, we determine the normality of the data distribution using the Shapiro–Wilk test [27]. If the Shapiro–Wilk test yields low statistical significance, suggesting approximate normality, we opt for an independent samples *t*-test. For the non-parametric analysis, we utilize the Mann–Whitney U test [28]. The rejection of the null hypothesis implies a significant difference in means or mean ranges (within a 95% confidence interval of the sample mean) between the two groups. In the case of parametric analysis, we apply the independent samples *t*-test to both equal variance and unequal variance groups. The distinction between the two is determined using Levene’s test for equality of variances [29]. An independent *t*-test was implemented to examine if there is statistical significant difference in mean values of all dimensions (“Family Interaction”, “Parenting”, “Emotional Well-being”, “Physical/Material Well-being”, “Disability-Related Support”; and “Authoritative Parenting Style”, “Authoritarian Parenting Style” and “Permissive Parenting Style”) before and after the intervention. 

In our study, we consider the intervention phase as the factor, specifically whether measurements were taken pre-intervention or post-intervention. A significant factor indicates an effect on both the central tendency and variability of the dependent variable. The direction of this effect (positive or negative) can be inferred from mean value plots based on the two intervention stages.

In cases where the observed results deviate from our expectations, either due to unexpected directions or a lack of statistical significance, we employ a cluster analysis methodology. This involves cluster analysis in two stages: First, we classify subjects as having either successful or unsuccessful interventions based on a new variable, “intervention”, indicating a success post-intervention and failure post-intervention. The second stage employs an algorithm to reclassify subjects into clusters based on their proximity to the cluster means. This enables the reclassification of subjects from “successful” to “unsuccessful” intervention clusters if their control variable values are closer to the “unsuccessful” cluster’s mean. 

In our study, the results of the analysis showed that there was an increase in all dimensions after the intervention, but they were not statistically significant at the 95% level of significance. That was our basic motivation: to examine cluster analysis for our sample (intervention group) in order to classify cases with a statistically significant improvement after the intervention. More precisely, in our analysis, we employed an embedded algorithm, offered through statistical software (SPSS 25.0), which classifies cases into clusters. The basic steps for the algorithm classification process were as follows: firstly, cases with common characteristics (in our case “dimensions’ results”) are selected, and secondly, cases in groups (here, we selected 2 groups) are classified using the Euclidean distance (there are many alternative metrics) between the examined cases and the previously created clusters as a metric. This algorithm resulted in the classification of the intervention group into 2 groups (70% of cases were successful, and 30% of cases were unsuccessful).

## 3. Results

### 3.1. Participants’ Profile 

Most of the participants were female (89.74%); 43.59% fell in the age range of 41–50, and 56.41% had two children. The majority of them were married (74.36%), high school graduates (53.85%), employed (64.10%) and reporting an income between EUR 10,001 and 15,000 (38.46%). More information is shown in Table 1.

### 3.2. Reliability of the Study Dimensions

The dimensions of the “quality of life” questionnaire (“Family Interaction”, “Parenting”, “Emotional Wellbeing”, “Physical/Material Wellbeing” and “Disability-Related Support”) and “parenting style” questionnaire (“Authoritative Parenting Style”, “Authoritarian Parenting Style” and “Permissive Parenting Style”) were subjected to reliability analysis with the Cronbach a coefficient ranging between 0.55 and 0.858 (see Table 2).

### 3.3. Effectiveness of the Intervention in Improving Parenting Style and Quality of Life

We performed baseline (pre-intervention) comparisons between the intervention and the control group. The results show that all the “Quality of life” dimensions (“Family Interaction”, “Parenting”, “Emotional Well-being”, “Physical/Material Well-being” and “Disability-Related Support”) had no statistically different mean values at baseline. In regard to the “Parenting style” dimensions, the results indicate that except for the “Authoritative Parenting Style”, the “Authoritarian Parenting Style” and the “Permissive Parenting Style” had statistically significant different mean values at baseline (see Table 3).

We also conducted post-intervention comparisons between the intervention and the control group (see Table 4). The analysis showed that all the “Quality of life” dimensions (“Family Interaction”, “Parenting”, “Emotional Well-being”, “Physical/Material Wellbeing” and “Disability-Related Support”) had different mean values for the two groups at a statistically significant level. In regard to the “Parenting styles”, both the “Authoritative” and the “Authoritarian” parenting styles had different mean values for the two groups, while the “Permissive Parenting Style” had no difference in mean values at post-intervention level. More precisely, a drop in the average value is observed at the post-intervention level for the variables “Authoritarian Parenting Style” and “Permissive Parenting Style”, while an increase is observed for the remaining variables, which, however, is relatively small.

According to the results of the intervention group, no variable showed a statistically significant difference in the mean values between the pre-intervention and the post-intervention scores greater than 95% (or *p*-value < 0.05) (see Table 5). 

According to the results presented in Table 6, the initial clusters considered as successful interventions, the cases that received the lowest possible values in the variables “Authoritarian Parenting Style” and “Permissive Parenting Style”, and the highest values in the remaining variables.

Unlike the initial clusters table, the final clusters presented in the table below take into account the actual mean values for most variables per intervention stage. Therefore, the analysis appears to be consistent with the descriptive statistics initially presented in the table of the characteristic elements of the sample of the experimental group (measures of position and dispersion). Therefore, the clustering should be considered successful.

According to the results of the ANOVA test (see Table 6), it appears that for all the variables, except for the “Authoritarian Parenting Style”, the intervention had an effect on their average values. Therefore, it should be checked whether the mean value is higher or lower for each variable after the intervention compared to the mean value before the intervention, although this is already established from the table regarding the final clusters.

The analysis (Table 6) further showed that “Family Interaction” significantly differs between the initial and final clusters, as indicated by a high F-statistic (20.71) and a very low *p*-value (0.000). Similarly, the “Parenting” variable exhibits significant differences between the initial and final clusters, with a high F-statistic (20.112) and a low *p*-value (0.000). The analysis demonstrates a substantial difference in “Emotional Well-being” between the initial and final clusters, with a remarkably high F-statistic (74.491) and a very low *p*-value (0.000). The variable “Physical/Material Well-being” also exhibits a significant difference between the initial and final clusters, with a high F-statistic (19.029) and a low *p*-value (0.000). The analysis showed a significant difference in “Disability-Related Support” between the initial and final clusters, with a high F-statistic (38.208) and a low *p*-value (0.000). “Authoritative Parenting Style” has a significant difference between the initial and final clusters, with a moderate F-statistic (5.574) and a *p*-value of 0.021, indicating a less significant difference compared to some other variables. The variable “Authoritarian Parenting Style” did not show a significant difference between the initial and final clusters, as the F-statistic is relatively low (2.34), and the *p*-value is higher than the conventional significance level of 0.05 (0.130). Similar to “Authoritarian Parenting Style”, “Permissive Parenting Style” does not exhibit a significant difference between the initial and final clusters, with a moderate F-statistic (40.187) and a very low *p*-value (0.000). The variable of “Authoritative Parenting Style” plus new dimension demonstrates a significant difference between the initial and final clusters, with a moderate F-statistic (9.421) and a *p*-value of 0.003.

In summary, the ANOVA results suggest that several factors related to “family interactions”, “parenting”, “emotional well-being”, “physical/material well-being” and “disability-related support” show significant differences between the initial and final clusters. This indicates that the intervention program had a notable impact on these aspects of the participants’ lives. However, “Authoritarian” and “Permissive Parenting Styles” did not significantly change between the clusters, while “Authoritative Parenting Style”, with a new, dimension did exhibit a significant difference.

## 4. Discussion

Summarizing the main findings, it appears that the intervention program introduced improvements in the parenting style and the quality of life dimensions of our participants but did not succeed in bringing a statistically significant change, as initially expected. Most importantly, these positive outcomes, although not statistically significant, occurred in the absence of other system changes, and this makes the current program even more successful. The major success of the intervention is that it achieved changes in the intervention group and not in the control group. Most importantly, the program had overwhelming acceptance and was regarded as beneficial to those who participated. Results though are mixed in terms of effectiveness, and further research is necessary to capture these changes in larger samples. 

In our study, we captured the small changes using a cluster analysis, and this offered important information and directions for future research. Specifically, a decrease in parental authoritarianism and an increase in support were observed, which is in accordance with Carroll [30], whose results suggest that participation in positive discipline parenting workshops led to noticeable changes in parenting styles over time. These changes encompassed reduced authoritarian and permissive tendencies, along with decreases in parental stress levels. Conversely, there was an increase in the adoption of a positive discipline style of parenting. Moreover, the findings of Gouveia et al. [31] imply that having greater dispositional mindfulness and self-compassion after an intervention may enhance the probability of embracing a mindful parenting approach in the parent–child relationship and support. This, subsequently, could be linked to the adoption of more effective parenting styles and a reduction in parenting-related stress.

Also, an increase in child–parent interactions as well as emotional and general well-being was observed. As the findings of Maddah’s et al.’s [32] study showed that the parental management training program had a significant positive impact. It effectively reduced excessive support and increased the parent care index among parents with children who have Sp.L.D.s. During the intervention, parents were coached on how to enhance their positive interactions with their children, while reducing conflicts and authoritarian interactions. Moreover, the study of Sahu et al. [33] introduced a promising and comprehensive intervention approach for Sp.L.D.s. This approach goes beyond solely addressing academic skill challenges and includes the emotional and behavioral well-being of parents and children with Sp.L.D.s. It also shifted the focus from center-based tutoring to home-based tutoring conducted by parents.

Certainly, the intervention program yielded positive outcomes, notably enhancing the quality of life (QoL) for the children involved in the study and their families. These findings align with a study conducted by Ginieri-Coccossis et al. [34], which suggested that Sp.L.D.s could significantly impact the QoL of newly diagnosed children. Moreover, the impact of the intervention extended to the parenting style and the overall QoL. This outcome resonates with Simon and Easvaradoss [35], who emphasized the irreplaceable influence parents exert on their children during their formative years. Additionally, the reduction in parental authoritarianism and the increase in support observed in our study align with Wellner’s [36] assertion that a trusting family–school relationship is especially crucial for parents of children with learning disabilities or behavioral problems. Furthermore, our results concerning the psychological aspects of parents align with Karande et al.’s [16] findings, which indicate that mothers of children with Sp.L.D.s tend to exhibit higher levels of anxiety compared to other groups of mothers. Additionally, our findings concerning the necessity and success of the intervention program align with a study conducted by Kausar et al. [37], which concluded that perceived social support plays a moderating role in the relationship between perceived stress and the QoL among parents of children with Sp.L.D.s. Moreover, our results further validate the importance of social support, as indicated in the study by Kuru and Piyal [38]. Their findings generate evidence in favor of the hypothesis that interventions and programs aimed at bolstering social support can greatly contribute to enhancing the lives of parents of children with Sp.L.D.s.

## 5. Study Limitations

This study has a number of limitations that need to be mentioned. Firstly, while the findings provide valuable insights into this specific area, they may not be fully representative of broader populations as the sample was derived from a population that used a community service under certain circumstances. Secondly, the tools employed in the current study were tested individually in past research but were used together for the first time here. Consequently, there are no comparable studies using this combination of tools. Third and of critical importance, the high drop-out rate in the control group may have affected the results of this study, as it did not allow for a more elaborate analysis and conclusions. The high rate of attrition has significantly reduced the amount of data and potentially the statistical power of analysis. If the subjects who dropped out differed from those who completed the intervention in terms of key study parameters, this may have significantly affected the outcome measures of this study, resulting in biased conclusions. 

## 6. Conclusions

The current study, despite not bringing a statistically significant change in participants’ parenting style and quality of life, found that the intervention program improved the parenting style of the participants, specifically by reducing authoritarianism and by increasing the interaction of parents with their children. It further became evident that the intervention program improved the QoL of the children by increasing all the dimensions of QoL. Most importantly, the current study offers evidence on the feasibility of the intervention in low-resource countries, its acceptability by people of various backgrounds including those with vulnerabilities, as well as its practicality for busy service providers and environments. 

## 7. Implications for Research and Clinical Practice

The findings of the current study can have multiple implications for research and practice. Firstly, we need to stress that more research is needed to explore the effectiveness of the intervention with larger samples of participants and with various research methods. Future studies may collect across multiple settings and assess parents prospectively, as part of longitudinal study designs. Future studies may also wish to better assess parents and explore other dynamic factors in depth (e.g., relationship quality and social supports) as well as better understand the parents’ adjustment and coping within different societies and social circumstances. Second, the current study offers a practical tool that may be used in different settings to improve parenting and family quality of life for children with Sp.L.D.s. The intervention may be used in health, education, justice and community settings as part of prevention and therapeutic sessions with people who need to prevent or even to improve negative behaviors in their children. It offers opportunities for brief counseling intervention even in busy non-clinical environments that may serve as points of reference for parents. In summary, parenting interventions have the potential to bring about positive changes at the individual, family and community levels. Parenting interventions can empower parents and increase their self-confidence and offer a sense of accomplishment. Children can experience a more nurturing and stable environment, which can positively impact their development, mental health, and overall well-being, and families as a whole can benefit from improved communication and reduced stress.

## Figures and Tables

**Figure 1 ijerph-20-07192-f001:**
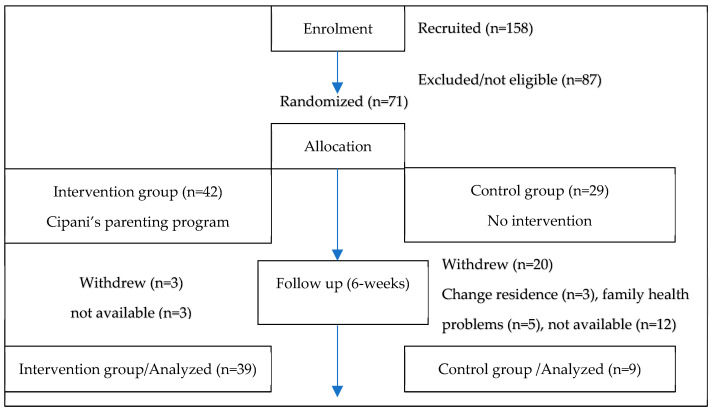
Randomized controlled trial flowchart.

**Table 1 ijerph-20-07192-t001:** Demographic profile of study participants.

	Intervention Group		Control Group	
Demographic Characteristics	Count	Percentage (%)	Count	Percentage (%)
Gender				
Female	35	90	27	93
Male	4	10	2	7
Age				
Up to 30	1	3	1	3
31–40	17	44	19	66
41–50	17	44	9	31
51 and above	4	10	0	0
Marital Status				
Single (Married)	4	10	1	3
Married	29	74	21	73
Divorced	4	10	7	24
Single Parent (Married)	1	3	0	0
Single Parent (Single)	1	3	0	0
Education				
Elementary School	2	5	1	3
Middle School	2	5	4	14
High School	21	54	13	45
Higher Education	10	26	9	31
Postgraduate Studies	4	10	2	7
Annual Family Income (Euros)				
Below 10,000	12	31	8	28
10,001–15,000	15	38	6	21
15,001–20,000	3	8	7	24
20,001–25,000	7	18	7	24
Over 25,001	1	3	1	3
No Response	1	3	0	0
Employment Status				
Employed	25	64	9	31
Unemployed	9	23	10	34
Self-employed	4	10	7	24
Homemaker	1	3	3	11
Number of Children				
1	1	3	2	7
2	22	56	16	55
3	10	26	7	24
4	5	13	2	7
No Response	1	3	2	7

**Table 2 ijerph-20-07192-t002:** Scale reliability analysis at pre- and post-intervention level.

DIMENSION	Pre-Intervention	Post-Intervention
Family Interaction	0.858	0.786
Parenting	0.844	0.723
Emotional Well-being	0.792	0.732
Physical/Material Well-being	0.764	0.488
Disability-related Support	0.806	0.663
Authoritative Parenting Style	0.796	0.783
Authoritarian Parenting Style	0.863	0.852
Permissive Parenting Style	0.544	0.641

**Table 3 ijerph-20-07192-t003:** Baseline (pre-intervention) comparisons between the intervention and the control group (independent *t*-test).

Dimensions	*p*-Value
Family Interaction	0.49
Parenting	0.26
Emotional Well-being	0.31
Physical Material Well-being	0.94
Disability-related Support	0.44
Authoritative Parenting Style	0.50
Authoritarian Parenting Style	0.00
Permissive Parenting Style	0.00

**Table 4 ijerph-20-07192-t004:** Post-intervention comparisons between the intervention and the control group (independent *t*-test).

	Group Statistics				
Dimensions	Group	N	Mean	Std. Deviation	*p*-Value
Family Interaction	intervention	39	4.47	0.47	0.00
	control	9	1.36	2.0	
Parenting	intervention	39	4.42	0.53	0.00
	control	9	1.4	2.1	
Emotional Well-being	intervention	39	3.66	0.92	0.00
	control	9	1.2	1.9	
Physical Material Well-being	intervention	39	4.65	0.42	0.00
	control	9	1.5	2.28	
Disability-related Support	intervention	39	4.51	0.52	0.00
	control	9	1.51	2.13	
Authoritative Parenting Style	intervention	39	4.58	0.33	0.00
	control	9	1.51	2.13	
Authoritarian Parenting Style	intervention	39	2.48	0.71	0.01
	control	9	1.49	2.11	
Permissive Parenting Style	intervention	39	1.94	0.74	0.19
	control	9	4.27	0.40	

**Table 5 ijerph-20-07192-t005:** Comparison between pre- and post-intervention performance in the intervention group (paired samples test).

		Sig. (Two-Tailed)
Pair 1	Family Interaction	0.075
Pair 2	Parenting	0.039
Pair 3	Emotional Well-being	0.140
Pair 4	Physical Material Well-being	0.239
Pair 5	Disability-related Support	0.096
Pair 6	Authoritative Parenting Style	0.155
Pair 7	Authoritarian Parenting Style	0.319
Pair 8	Permissive Parenting Style	0.202

**Table 6 ijerph-20-07192-t006:** Results of ANOVA control with factor separation in the new clusters.

	Initial Clusters	Final Clusters	Results of ANOVA Control with Factor Separation in the New Clusters					
	1	2	1	2	Mean Square Error	Df	F	Sig.
Family Interaction	1.67	5.00	4.03	4.60	0.261	74	20.71	0.000
Parenting	1.50	5.00	3.91	4.55	0.341	74	20.112	0.000
Emotional Well-being	1.75	5.00	2.64	4.08	0.469	74	74.491	0.000
Physical/Material Well-being	2.80	5.00	4.30	4.80	0.197	74	19.029	0.000
Disability-related Support	1.00	5.00	3.86	4.73	0.333	74	38.208	0.000
Authoritative Parenting Style	4.77	5.00	4.44	4.63	0.115	74	5.574	0.021
Authoritarian Parenting Style	2.08	1.17	2.80	2.52	0.584	74	2.34	0.130
Permissive Parenting Style	2.50	1.00	2.74	1.76	0.401	74	40.187	0.000

## Data Availability

The data that support the findings of this study are available from the corresponding author upon reasonable request.

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
