# Peer review of "Design, Implementation and Evaluation of an Innovative Pilot Intervention to Improve the Family Quality of Life of Children with Specific Learning Disabilities—A Randomized Controlled Trial"

_ijerph, 2023, doi:10.3390/ijerph20247192_

Round 1
Reviewer 1 Report
Comments and Suggestions for Authors
This is a report of a RCT in which parents of children with specific learning difficulties were assigned to a parent training intervention group or control group. Outcomes were parenting style and family quality of life, both assessing using a parent-report questionnaire.
The paper addresses important issues, but doesn’t yet provide the detail and clarity required for publication.
TITLE
Please indicate that this is a randomized controlled trial. You don’t need to say “a quantitative analysis”.
ABSTRACT
Please mention the research design (randomized controlled trial), population (parents of children with specific learning difficulties), when the outcomes were assessed, the names of the measures used, and main statistical results.
INTRODUCTION
The second paragraph of the Introduction is a startlingly harsh description of parents of children with learning difficulties. This needs to be substantially rewritten in a more balanced way. When you have done this, it would be good to ask one or more parents of a child with learning difficulties to review this paragraph to make sure they consider that it is a fair description, and that is respectful.
The last 2 paragraphs of the Introduction give a lot of aims and research questions. After reading the remainder of the paper, I think it boils down to 2: Does the intervention improve parenting style? and does it improve family quality of life? Could these 2 paragraphs be simplified?
METHODS
I will be asking for much more detail here:
1. Study design is a randomized controlled trial. Please state this. And presumably parallel groups?
2. Is there a published protocol?
3. What Ethics approval was obtained?
4. In what settings were the data collected? Face-to-face? Video-call? In participants’ home? In a clinic?
5. Please give enough detail about the intervention to allow others to replicate it. You may need to add supplementary material to the paper for this.
6. Who delivered the intervention? Was it all delivered individually? Or were some individual and some group-based?
7. How much time was spent in the 4 meetings? How much time did families spend between meetings?
8. Did you assess adherence to the intervention program? How?
9. What did the control group do and how much time did they spend doing it?
10. How was sample size determined?
11. What method was used to assign participants randomly to groups?
12. Was there random allocation concealment? And if so, how was this done?
13. Was there any blinding? Who was blinded? Presumably not those who delivered the intervention or the parents, but were assessors blinded to group allocation? Was the data analyst blinded?
14. At what point were the outcomes assessed? I think it was just before the intervention and straight after? Please state this explicitly. Was there any follow-up?
DATA ANALYSIS
The data analysis section is complicated and I think could be simplified. You want to know whether the intervention group did better than the control group. So, I would expect to see a comparison of the 2 groups at baseline, giving both descriptives and confidence intervals or p-values (from an independent samples t-tests or Mann-Whitney). Then either the same at post-test, or an ANCOVA, adjusting for the pre-test scores (again with descriptives).
As it stands, the data analysis section says that “dimensions of the quality of life and parental style questionnaires were calculated” but no dimensions were mentioned in the Methods, which gave the impression that there were only 2 outcome measures. Are there subscales on these 2 questionnaires, and if there are, please state them in the Methods and carry them through to the Data Analysis and Results.
The data analysis talks about “exclusion from the sample of cases that did not follow the standards of correct completion”. Please could you indicate at the beginning of the Results whether you excluded anyone from the study, at what point and why. (I’ll repeat this in the Results below.)
The data analysis says that reliability results were calculated, but these are not reported in the Results.
The 3 bullet points are not specific enough. What was the multivariate analysis? The last 2 bullet points seem to be saying the same thing “Comparisons of “quality of life” and “parental style” scores were implemented, before and after the intervention”. And neither of them indicates the analysis used.
The next paragraph distinguishes between independent t-tests (which were used to compare the groups with one another) and ANOVA (which was used to compare pre-intervention with post-intervention time points). If you wish to compare these, why wouldn’t you use a paired t-test (or a Wilcoxon signed ranks for non-parametric)?
I find the description of the cluster analysis unclear. It says “we classify subjects as having either successful or unsuccessful interventions based on a new variable, "intervention"”. How was it determined whether the intervention was successful or not? And does this mean that the cluster analysis was done only on the participants in the intervention group? I’m afraid I can’t follow the second stage of the cluster analysis at all. What was the algorithm? How was it derived? And what are “their control variable values”?
RESULTS
Please specify how many were recruited, whether any were dropped, at what point and what the reasons were. A flowchart might help here. The Methods said that “the participants were randomly divided into two groups (29 in the control group and 39 in the intervention group).” So there were 29 + 39 = 68 participants recruited, but Table 1 gives data for only 35. Does that mean that 33 participants left the study? When and why? Or are the 35 participants all from the intervention group? In which case, what happened to the other 4? And what are the results for the control group?
In Table 1, please provide data for the 2 groups at baseline, not all the participants together. This will allow readers to assess how similar they were at baseline. For this sample size, you don’t need to give any decimal places (e.g., 89.74% can be rounded to 90%).
In Table 2, please provide descriptive data for the 2 groups at post-test, such as mean, SD (or, if non-parametric statistics are used, median and IQR). The text says that descriptive stats are presented in Table 2, but they are not.
The text says that “t-test or Mann-Whitney test” was used to generate the results in Table 2. But Table 2 shows analysis of pre versus post intervention, so these would not be appropriate. Independent t-tests and Mann-Whitney tests should be used to compare the groups with one another, not the same participants at different time points.
In reporting p-values, please use decimal points (.) not commas (,).
The paper doesn’t report the analysis that I would expect, namely comparisons between groups on the outcome measures to see whether the intervention group improved more than the control group.
Could the cluster analysis be clarified much more? I find difficult to reconcile the analysis in Table 2 (which found no effects) with the cluster analysis (which found extremely strong and consistent effects).
DISCUSSION
I’m going to leave the Discussion at this stage. The answers to my questions above regarding methods, data analyses, and results may lead to a change of results. At any rate, they will make them clearer and more interpretable to me. And I need to understand them properly before I can review the Discussion.
Comments on the Quality of English LanguageEnglish is fine.
Author Response
|
Response to Reviewer 1 Comments
|
||||||||||||||||||||||||||||||||||||||||
|
Summary
|
|
|
||||||||||||||||||||||||||||||||||||||
|
Thank you very much for taking the time to review our manuscript. Many thanks for your comments and your contribution to our research. Please find the detailed responses below and the corresponding revisions/corrections highlighted/in track changes in the re-submitted files.
|
||||||||||||||||||||||||||||||||||||||||
|
Point-by-point response to Comments and Suggestions for Authors
|
||||||||||||||||||||||||||||||||||||||||
|
Title Comments 1: Please indicate that this is a randomized controlled trial. You don’t need to say “a quantitative analysis”. Response 1: Thank you for this comment. The title has been changed and now reads as follows: “Design, implementation and evaluation of an innovative pilot intervention to improve the quality of life of the family of students with specific learning disabilities. A randomized controlled trial”.
Abstract Comments 2: Please mention the research design (randomized controlled trial), population (parents of children with specific learning difficulties), when the outcomes were assessed, the names of the measures used, and main statistical results. Response 2: Abstract: Background: Many thanks for this point. We have introduced all the necessary changes in the abstract.
Introduction Comments 3: The second paragraph of the Introduction is a startlingly harsh description of parents of children with learning difficulties. This needs to be substantially rewritten in a more balanced way. When you have done this, it would be good to ask one or more parents of a child with learning difficulties to review this paragraph to make sure they consider that it is a fair description, and that is respectful. Response 3: The reviewer is raising a very crucial point and we very much appreciate this comment. Previous research was included unfiltered. We have now reviewed and rephrased the content of the paragraph.
Comments 4: The last 2 paragraphs of the Introduction give a lot of aims and research questions. After reading the remainder of the paper, I think it boils down to 2: Does the intervention improve parenting style? and does it improve family quality of life? Could these 2 paragraphs be simplified? Response 4: Many thanks for this comment. The text was rephrased according to the reviewer suggestion, to clearly define the objectives and research questions of the study.
Methods Comments 5: Study design is a randomized controlled trial. Please state this. And presumably parallel groups? |
||||||||||||||||||||||||||||||||||||||||
|
Response 5: Thank you for pointing this out. It was a randomized control trial with intervention and control group. The study design is now described in detail under the “Material and Methods section”.
|
||||||||||||||||||||||||||||||||||||||||
|
Comments 6: Is there a published protocol? |
||||||||||||||||||||||||||||||||||||||||
|
Response 6: The study protocol was not published.
Comments 7: What Ethics approval was obtained? Response 7: Approval was granted by the Ethics Committee of the Hellenic Mediterranean University. Reference number is included at the end of the manuscript.
Comments 8: In what settings were the data collected? Face-to-face? Video-call? In participants’ home? In a clinic? Response 8: Thank you once again for this comment. Indeed, different means of communication were used for the purposes of the intervention, as follows: 1st session was face-to-face while the 2nd 3rd and 4th sessions used video-call. This is now clearly stated under the Material and Methods section and specifically under the “2.4. Content of the intervention”
Comments 9: Please give enough detail about the intervention to allow others to replicate it. You may need to add supplementary material to the paper for this. Response 9: Thank you for pointing this out. We agree with this comment. Therefore, we have described the intervention in detail under the “Material and Methods section”.
Comments 10: Who delivered the intervention? Was it all delivered individually? Or were some individual and some group-based? Response 10: The intervention was conducted by researchers/mental health professionals with a background in social work or psychology and long experience in the field of special education and family counseling. All interventions were individual. All the information is now provided in detail under the “Material and Methods section” and specifically under the “2.4. Content of the intervention”.
Comments 11: How much time was spent in the 4 meetings? How much time did families spend between meetings? Response 11: Each session lasted 2 hours and they were completed in a 6-week period. All the information is now provided in detail under the “Material and Methods section” and specifically under the “2.4. Content of the intervention”.
Comments 12: Did you assess adherence to the intervention program? How? Response 12: Adherence to the study as “the percentage of individuals completing the intervention against those lost in follow up” is now reported under the “Material and Methods section” and specifically under the “2.2. Participants’ recruitment”.
Comments 13: What did the control group do and how much time did they spend doing it? Response 13: The control group received no intervention and individuals were included in a waitlist to have access to the program at a later stage. This is now indicated in text under the “Material and Methods section” and specifically under the “2.2. Participants’ recruitment”.
Comments 14: How was sample size determined? Response 14: All the people treated by the university-led mobile unit, that fulfilled the eligibility criteria, were randomized. A detailed description of the randomization process is now included under the “Material and Methods section” and specifically under the “2.2. Participants’ recruitment”.
Comments 15: What method was used to assign participants randomly to groups? Response 15: Thank you for indicating this missing point. Computer-assisted methods were used. A detailed description of the randomization process is now included under the “Material and Methods section” and specifically under the “2.2. Participants’ recruitment”.
Comments 16: Was there random allocation concealment? And if so, how was this done? Response 16: Concealment was managed by engaging independent researchers in the randomization process and by using computer-generated lists. Details are included under the “Material and Methods section” and specifically under the “2.2. Participants’ recruitment”.
Comments 17: Was there any blinding? Who was blinded? Presumably not those who delivered the intervention or the parents, but were assessors blinded to group allocation? Was the data analyst blinded? Response 17: Yes indeed. External experts reviewing the results of the intervention, were blinded. This is clearly stated under the “Material and Methods section” and specifically under the “2.2. Participants’ recruitment”.
Comments 18: At what point were the outcomes assessed? I think it was just before the intervention and straight after? Please state this explicitly. Was there any follow-up? Response 18: Yes, indeed, there has been a baseline and a post-intervention evaluation, during the month following completion. Annual follow-up has been planned. This is clearly stated under the “Material and Methods section” and specifically under the “2.1. Study design”.
Data analysis Comments 19: The data analysis section is complicated and I think could be simplified. You want to know whether the intervention group did better than the control group. So, I would expect to see a comparison of the 2 groups at baseline, giving both descriptives and confidence intervals or p-values (from an independent samples t-tests or Mann-Whitney). Then either the same at post-test, or an ANCOVA, adjusting for the pre-test scores (again with descriptives). Response 19: We agree with this comment. Thank you for pointing this out. We agree with this comment. Table 2 has been updated to include all the necessary information. This applies also to comment 28.
Comments 20: As it stands, the data analysis section says that “dimensions of the quality of life and parental style questionnaires were calculated”, but no dimensions were mentioned in the Methods, which gave the impression that there were only 2 outcome measures. Are there subscales on these 2 questionnaires, and if there are, please state them in the Methods and carry them through to the Data Analysis and Results. Response 20: Thank you for pointing this out. We have now mentioned all the “Parenting style” and “quality of life” dimensions under the Material and Methods section. They were previously mentioned only under the Results section.
Comments 21: The data analysis talks about “exclusion from the sample of cases that did not follow the standards of correct completion”. Please could you indicate at the beginning of the Results whether you excluded anyone from the study, at what point and why (I’ll repeat this in the Results below). Response 21: Thank you for pointing this out. We have included all the information in detail in the flow chart together with description in text.
Comments 22: The data analysis says that reliability results were calculated, but these are not reported in the Results. Response 22: Thank you for indicating this point. We have included a new table (Table 1) presenting Cronbach Reliability Analysis
Comments 23: The 3 bullet points are not specific enough. What was the multivariate analysis? The last 2 bullet points seem to be saying the same thing “Comparisons of “quality of life” and “parental style” scores were implemented, before and after the intervention”. And neither of them indicates the analysis used. Response 23: Thank you for pointing this out. Multivariate analysis was implemented for educational and academic proposes, but it was not a part of this paper discussion. It was deleted. We have changed bullet points as follows: · Comparisons of dimensions of “quality of life” and dimensions of “parental style” scores were implemented, before and after the intervention mainly for the intervention group. We used independent t-test and (as it has been noticed) the most suitable paired t-tests. · Comparisons of “quality of life” and “parental style” scores were implemented mainly before the intervention, between the intervention and the control group. We used independent t-test. There were many withdrawals from the second test from control group, and the results after the intervention were at least unstable for statistical conclusions.
Comments 24: The next paragraph distinguishes between independent t-tests (which were used to compare the groups with one another) and ANOVA (which was used to compare pre-intervention with post-intervention time points). If you wish to compare these, why wouldn’t you use a paired t-test (or a Wilcoxon signed ranks for non-parametric)? Response 24: Thank you for pointing this out. It was very helpful for us to further improve our description and analytical methods. All these issues have been clarified in text. Basically we calculated both (t-tests and paired test) for the intervention group. Independent t- test was implemented to examine if there is statistical significant difference in mean values of all dimensions (Family Interaction, Parenting, Emotional Well-being, Physical/Material Well-being,, Disability-Related Support and Authoritative Parenting Style, Authoritarian Parenting Style, Permissive Parenting Style) before and after the intervention. Moreover, we performed a paired t-test to examine if there was a statistically significant difference in all dimensions for each case, for the intervention group. The results of the analysis showed that there was an increase in all dimensions after the intervention, but not statistical significant at 95% level of significance. That was our basic motivation to examine cluster analysis for our sample (intervention group) in order to classify cases with statistical significant improvement after the intervention.
Comments 25: I find the description of the cluster analysis unclear. It says “we classify subjects as having either successful or unsuccessful interventions based on a new variable, "intervention"”. How was it determined whether the intervention was successful or not? And does this mean that the cluster analysis was done only on the participants in the intervention group? I’m afraid I can’t follow the second stage of the cluster analysis at all. What was the algorithm? How was it derived? And what are “their control variable values”? Response 25: We agree. Many thanks. Therefore, we have clarified the study methods, in detail, under the “Results section”.
Results Comments 26: Please specify how many were recruited, whether any were dropped, at what point and what the reasons were. A flowchart might help here. Response 26: Thank you for raising this important point. This is now clearly stated under the “Material and Methods section” and specifically under the “2.2. Participants’ recruitment”. A flow chart has been added to facilitate understanding of the recruitment process.
Comments 27: The Methods said that “the participants were randomly divided into two groups (29 in the control group and 39 in the intervention group).” So there were 29 + 39 = 68 participants recruited, but Table 1 gives data for only 35. Does that mean that 33 participants left the study? When and why? Or are the 35 participants all from the intervention group? In which case, what happened to the other 4? And what are the results for the control group? Response 27: This is an important comment and similar to Comment 26. This is now clear under the “2.2. Participants’ recruitment”.
Comments 28: In Table 1, please provide data for the 2 groups at baseline, not all the participants together. This will allow readers to assess how similar they were at baseline. For this sample size, you don’t need to give any decimal places (e.g., 89.74% can be rounded to 90%). Response 28: Thank you for pointing this out. We agree with this comment. Table 2 has been updated to include all the necessary information.
Comments 29: In Table 2, please provide descriptive data for the 2 groups at post-test, such as mean, SD (or, if non-parametric statistics are used, median and IQR). The text says that descriptive stats are presented in Table 2, but they are not. Response 29: Thank you for pointing this out. We agree with this comment. Therefore, this is a new table (included as Table 3) for the abovementioned requirements.
Comments 30: The text says that “t-test or Mann-Whitney test” was used to generate the results in Table 2. But Table 2 shows analysis of pre versus post intervention, so these would not be appropriate. Independent t-tests and Mann-Whitney tests should be used to compare the groups with one another, not the same participants at different time points. Response 30: Thank you for pointing this out. We agree with this comment. We performed and paired t-test analysis too, which is included as new table 4.
Comments 31: In reporting p-values, please use decimal points (.) not commas (,). Response 31: Ok, thank you. We have changed it.
Comments 32: The paper doesn’t report the analysis that I would expect, namely comparisons between groups on the outcome measures to see whether the intervention group improved more than the control group. Response 32: Thank you for pointing this out. We agree with this comment. Therefore, are the updated results: Baseline comparison We did baseline (phase 1) comparisons between intervention and control group. The Results shows that dimensions Family Interaction, Parenting, Emotional Wellbeing, Physical Material Wellbeing, Disability Related Support and Authoritative Parenting Style have the same level of mean values for the 2 groups. On the other hand, Authoritarian Parenting Style and Permissive Parenting Style have statistical different mean values at baseline.
Phase 2 comparison
There were only 9 cases in phase 2 from control group. The results are represented above but there no statistical evidence.
We made phase 2 comparisons between intervention and control group. The Results shows that dimensions Family Interaction, Parenting, Emotional Wellbeing, Physical Material Wellbeing, Disability Related Support and Authoritative Parenting Style have had statistical different mean values for the 2 groups. Only Permissive Parenting Style have no difference in mean values at phase 2.
Comments 33: Could the cluster analysis be clarified much more? I find difficult to reconcile the analysis in Table 2 (which found no effects) with the cluster analysis (which found extremely strong and consistent effects). Response 33: Thank you for pointing this out. We agree with this comment. Therefore, we have clarified that was the result of the classification of intervention group in 2 groups (70% of cases successfully in success group and 30% cases unsuccessful).
|
||||||||||||||||||||||||||||||||||||||||
Reviewer 2 Report
Comments and Suggestions for Authors
This is a research about improve the quality of life of the family of students with specific learning disabilities. This topic is very interesting, well written and well structured. However, there are some problems that need to be revised by the authors.
The first is the key point. I'm not sure if ijerph needs to provide this part, because I haven't seen it in other articles published in this journal.
The second is the introduction. I think the introduction is very lengthy, but it doesn't focus. I suggest that the authors divide it into two parts, one is introduction, the other is literature review and research questions.
The third is research tools. It seems that the analysis methods of output variables and control variables are not used in this study, but experimental intervention is adopted. Then, the authors' expression of this part needs to be revised according to the requirements of the journal.
The fourth is the research results. It's obviously not enough for you to just use ANOVA's analysis, so it's difficult to directly draw the conclusion that the experiment is effective. At the same time, your demographic variables are only described in general, without any control or related analysis. This part needs to be revised by the authors.
Finally, I think this is a good study, but the authors need to make appropriate revisions to ensure that it meets the publishing requirements.
Author Response
|
Response to Reviewer 2 Comments
|
||
|
Summary
|
|
|
|
Thank you very much for taking the time to review this manuscript. Please find the detailed responses below and the corresponding revisions/corrections highlighted/in track changes in the re-submitted files. We also thank you for your contribution and we agree with all your comments.
Point-by-point response to Comments and Suggestions for Authors
|
||
|
|
|
|
|
Comments 1: The first is the key point. I'm not sure if ijerph needs to provide this part, because I haven't seen it in other articles published in this journal. Response 1: Thank you for pointing this out. We agree with this comment. Therefore, we have deleted it.
Comments 2: The second is the introduction. I think the introduction is very lengthy, but it doesn't focus. I suggest that the authors divide it into two parts, one is introduction, the other is literature review and research questions. Response 2: Thank you for this useful comment. We have, adapted the introduction according to the reviewer’s suggestion.
Comments 3: The third is research tools. It seems that the analysis methods of output variables and control variables are not used in this study, but experimental intervention is adopted. Then, the authors' expression of this part needs to be revised according to the requirements of the journal. Response 3: Many thanks for the comment. The specific part has been revised accordingly.
Comments 4: The fourth is the research results. It's obviously not enough for you to just use ANOVA's analysis, so it's difficult to directly draw the conclusion that the experiment is effective. At the same time, your demographic variables are only described in general, without any control or related analysis. This part needs to be revised by the authors. Response 4: New tables have been included and new analytical methods were used during the revision process, according to the reviewers’ suggestions. The analysis is now more clear.
|
||
|
|
||
|
|
||

Round 2
Reviewer 1 Report
Comments and Suggestions for Authors
This is the second round of review for an RCT of parents of children with learning difficulties, who received an intervention or control. Outcomes were parenting style and family quality of life, assessed by parent-completed questionnaire.
In my previous report, I said that although the article paper addresses important issues, it didn’t provide enough detail and clarity to allow it to be accepted for publication. The authors have made considerable changes and have addressed many of my concerns. However, I still have a number of queries. Importantly, I am not convinced that the conclusion is justified from the data.
ABSTRACT
The authors have made some revisions in response to my comments, thank you.
But could you also please add when the outcomes were assessed (e.g., pre-intervention, post-intervention, and whether there was a follow-up) and give statistical results?
KEY POINTS
The key points of this article have been deleted but not replaced.
INTRODUCTION
This is considerably improved and much clearer, thank you.
The aims are simplified. However, I do not think the paper addresses Aims 1 and 2, but only Aim 3.
METHODS
This contains much more detail, and answers many of my queries, thank you.
SETTINGS: I asked last time, “In what settings were the data collected? Face-to-face? Video-call? In participants’ home? In a clinic?” You responded by explaining the settings for the intervention. This is important information, thank you. But I was really asking about data collection, that is, the administration of the questionnaires to participants in both groups at pre-test and post-test. Please mention the settings in which this was done.
The word “missions” is sometimes used where I think it should say “sessions”.
RANDOMIZATION: Thank you for explaining the process of randomization in detail. Could you also please indicate whether you had “allocation concealment”? In other words, whether the person who decided whether each participant met inclusion criteria (entered the study) knew which group each participant would be assigned to.
When you say, “The list was then resorted to its original order”, I don’t quite understand. Could you rephrase this more clearly?
ADHERENCE: Last time, I asked whether you assessed adherence to the intervention program. You have provided information about drop-outs, which very important, thank you. But I meant: did you collect information about the number sessions attended or completion of homework between sessions?
BLINDING: You mention that 2 researchers were blinded and they reviewed the results. Were the researchers who administered the questionnaires to the parents (the assessors) blinded? And was the data analyst blinded?
DATA ANALYSIS
OUTCOME MEASURES: In Section 2.3, there are 2 outcome measures (parenting style and family QOL). But in Section 2.5, there are 8 outcome measures (“Family Interaction”, “Parenting”, “Emotional Wellbeing”, “Physical/Material Wellbeing”, “Disability-Related Support”, “Authoritative Parenting Style”, “Authoritarian Parenting Style”, Permissive Parenting Style). I would guess that the first 5 (“Family Interaction”, “Parenting”, “Emotional Wellbeing”, “Physical/Material Wellbeing”, “Disability-Related Support”) come from the family QOL questionnaire, and the last 3 (“Authoritative Parenting Style”, “Authoritarian Parenting Style”, Permissive Parenting Style) come from the parenting style questionnaire. Could you please specify in Section 2.3, what the subscales of each of the 2 measures are? And could you also please explain what higher and lower values mean on these subscales?
SCALE RELIABILITY RESULTS: Please move Table 1 to the Results.
MAIN ANALYSES: I’m afraid these analyses are still not clearly laid out. There is repetition and sometimes apparent self-contradiction.
The first bullet point seems to be a comparison between the results before and after intervention. I would expect a paired t-test for this comparison. But the text mentions both independent t-test and paired t-tests for this analysis.
The second bullet point is a comparison between the groups before the intervention (which rightly uses an independent t-test). But there is no comparison between the groups AFTER the intervention. This is the most important comparison for an RCT. Later on, you say that “independent t-test was implemented to examine if there is statistical significant difference in mean values of all dimensions… before and after the intervention.” So evidently, the post-intervention results WERE analysed using independent t-tests.
The next sentence (“Moreover, paired t-test was performed to examine if there was a statistically significant difference in all dimensions for each case, for the intervention group.”) seems to duplicate the first bullet point.
Within these bullet points on the main analyses, also, should be mentioned the possibility of using the non-parametric equivalents of independent t-tests and paired t-tests, in case the parametric assumptions are not met. These are only mentioned later on in the data analysis section.
RESULTS
Table 2 appears twice in my copy.
COMPARISON AT POST-INTERVENTION: Table 3 shows the results at post-intervention for the 2 groups clearly, thank you. Could you please indicate which were statistically significantly different?
COMPARISON AT PRE-INTERVENTION: Could you please add another table similar to Table 3, showing the results pre-intervention. From the answer to my question in the “response to authors”, I see that there were no differences in any of the outcome measures pre-intervention except for authoritarian and permissive parenting styles. This would be useful for readers to see.
PHASE 2 or B PHASE: In the methods, “phase 2” referred to the last 3 sessions of the intervention. However, the results seem to use “phase 2” and “B phase” to refer to post-intervention results. Am I understanding this correctly? It would be clearer to use post-intervention throughout, when referring to the questionnaire results taken at the end of the study (rather than Phase 2 or B phase).
TABLE 4: Please explain in text what “Authoritative Parenting Style plus new dim - Β phase Authoritative Parenting Style plus new dim” is.
PHASE 2 T-TEST RESULTS: In your response to reviewer (but not in the paper), you include 2 tables of t-tests, one at baseline and one at Phase 2 (which I take to be post-intervention). These are good, thank you. I’m puzzled though, because the second of those tables shows significant differences for all outcomes post-intervention, except permissive parenting style. But Table 3 in your paper seems to show bigger differences between the groups in authoritarian and permissive parenting styles than in any of the other variables. Just looking at the Table 3 results, I would have guessed that there might be significant differences between groups on authoritarian and permissive parenting styles, but certainly not on any of the other outcomes. So the t-test results don’t appear to match the descriptive data. Am I misunderstanding something here?
CLUSTER ANALYSIS: I still cannot understand what this analysis contributes to the paper. From the results so far, it appears to me that the intervention might have had an effect on authoritarian and permissive parenting styles (in Table 3), but not on any of the other outcomes (in Tables 3 and 4). However, the conclusion from the cluster analysis is that “the intervention program had a notable impact on these aspects of the participants' lives”. How could these 2 results both be true? They are inconsistent with one another.
Moreover, the abstract presents only this conclusion (not the results from the traditional analysis, showing little or no difference on most of the outcome measures). This conclusion doesn’t seem to be sufficiently justified from the data.
DISCUSSION
The Discussion is based on the conclusion drawn from the cluster analysis, and I’m not convinced that this is the correct conclusion from the data that has been presented.
The limitations need to place more weight on the level of attrition. It not only limited the analyses. It probably also resulted in a more biased sample, because participants do not drop out randomly. Two thirds of the control group dropped out. This is an extremely high rate, especially for a group who didn’t have to anything except complete 2 questionnaires at the end of 6 weeks. It deserves serious consideration.
Comments on the Quality of English Language
Only minor editing of English required
Author Response
Thank you very much. Your comments have been very important and contributed to the improvement of our manuscript. Our responses have been included with track-changes in the revised manuscript. Below, you will find a point-by-point response.
Comment 1
This is the second round of review for an RCT of parents of children with learning difficulties, who received an intervention or control. Outcomes were parenting style and family quality of life, assessed by parent-completed questionnaire.
In my previous report, I said that although the article paper addresses important issues, it didn’t provide enough detail and clarity to allow it to be accepted for publication. The authors have made considerable changes and have addressed many of my concerns. However, I still have a number of queries. Importantly, I am not convinced that the conclusion is justified from the data.
Response 1
Thank you for your kind remarks. We have tried to address the rest of the comments raised by the reviewer.
Comment 2
ABSTRACT
The authors have made some revisions in response to my comments, thank you. But could you also please add when the outcomes were assessed (e.g., pre-intervention, post-intervention, and whether there was a follow-up) and give statistical results?
Response 2
We have re-written the abstract to better present the methods-results and conclusions of the study. Thank you.
Comment 3
KEY POINTS
The key points of this article have been deleted but not replaced.
Response 3
The key points were removed upon the second reviewer’s request because they were not in line with the journal’s guidelines.
Comment 4
INTRODUCTION
This is considerably improved and much clearer, thank you.
Response 4
Thank you.
Comment 5
The aims are simplified. However, I do not think the paper addresses Aims 1 and 2, but only Aim 3.
Response 5
The reviewer is right. The current manuscript reports on the third objective. This is now corrected under the paragraph “1.2 Research goals and questions”. Thank you.
Comment 6
METHODS
This contains much more detail, and answers many of my queries, thank you.
Response 6
Thank you.
Comment 7
SETTINGS: I asked last time, “In what settings were the data collected? Face-to-face? Video-call? In participants’ home? In a clinic?” You responded by explaining the settings for the intervention. This is important information, thank you. But I was really asking about data collection, that is, the administration of the questionnaires to participants in both groups at pre-test and post-test. Please mention the settings in which this was done.
Response 7
Thank you very much for the comment. Data collection was carried out via online forms that were distributed by certain members of the research team, in charge of this task. This is now mentioned under the paragraph “2.3. Outcome measures”.
Comment 8
The word “missions” is sometimes used where I think it should say “sessions”.
Response 8
Thank you once again for this point. Indeed, the term “missions” was not appropriate. We referred to the “home-based assignments”, which were part of the intervention. This is now corrected at two points under the paragraph “2.4. Content of the Intervention”.
Comment 9
RANDOMIZATION: Thank you for explaining the process of randomization in detail. Could you also please indicate whether you had “allocation concealment”? In other words, whether the person who decided whether each participant met inclusion criteria (entered the study) knew which group each participant would be assigned to.
When you say, “The list was then resorted to its original order”, I don’t quite understand. Could you rephrase this more clearly?
Response 9
Indeed, eligibility check was a separate process. Many thanks for this important comment. This is now clear under the paragraph “2.2. Participants’ recruitment”.
Comment 10
ADHERENCE: Last time, I asked whether you assessed adherence to the intervention program. You have provided information about drop-outs, which very important, thank you. But I meant: did you collect information about the number sessions attended or completion of homework between sessions?
Response 10
Many thanks for the clarification. None of the sessions were missed by our intervention group. It was a short intervention and none missing sessions were recorded except from the two drop-outs.
Comment 11
BLINDING: You mention that 2 researchers were blinded and they reviewed the results. Were the researchers who administered the questionnaires to the parents (the assessors) blinded? And was the data analyst blinded?
Response 11
Indeed, independent researchers were assigned with questionnaire administration, data collection and data analysis. This is now made clear under the paragraph “2.2. Participants’ recruitment”.
Comment 12
DATA ANALYSIS
OUTCOME MEASURES: In Section 2.3, there are 2 outcome measures (parenting style and family QOL). But in Section 2.5, there are 8 outcome measures (“Family Interaction”, “Parenting”, “Emotional Wellbeing”, “Physical/Material Wellbeing”, “Disability-Related Support”, “Authoritative Parenting Style”, “Authoritarian Parenting Style”, Permissive Parenting Style). I would guess that the first 5 (“Family Interaction”, “Parenting”, “Emotional Wellbeing”, “Physical/Material Wellbeing”, “Disability-Related Support”) come from the family QOL questionnaire, and the last 3 (“Authoritative Parenting Style”, “Authoritarian Parenting Style”, Permissive Parenting Style) come from the parenting style questionnaire. Could you please specify in Section 2.3, what the subscales of each of the 2 measures are? And could you also please explain what higher and lower values mean on these subscales?
Response 12
Thank you very much for this critical comment. The questionnaire dimensions and their components are now clearly presented under the paragraphs “2.3. Outcome Measures” and “2.5. Statistical analysis”.
Comment 13
SCALE RELIABILITY RESULTS: Please move Table 1 to the Results.
Response 13
We have done it. Thank you.
Comment 14
MAIN ANALYSES: I’m afraid these analyses are still not clearly laid out. There is repetition and sometimes apparent self-contradiction.
The first bullet point seems to be a comparison between the results before and after intervention. I would expect a paired t-test for this comparison. But the text mentions both independent t-test and paired t-tests for this analysis.
Response 14
Thank you very much for this comment. This has now been clarified under the paragraph “2.5. Statistical analysis”. Paired t-test is the only analysis applied for this comparison. Independent t-test was included by mistake and has now been removed.
Comment 15
The second bullet point is a comparison between the groups before the intervention (which rightly uses an independent t-test). But there is no comparison between the groups AFTER the intervention. This is the most important comparison for an RCT. Later on, you say that “independent t-test was implemented to examine if there is statistical significant difference in mean values of all dimensions… before and after the intervention.” So evidently, the post-intervention results WERE analysed using independent t-tests.
Response 15
The reviewer is correct. Thank you for the comment. This information is now included in Table 4 together with explanations.
Comment 16
The next sentence (“Moreover, paired t-test was performed to examine if there was a statistically significant difference in all dimensions for each case, for the intervention group.”) seems to duplicate the first bullet point.
Response 16
Thank you for indicating this duplication. The sentence was removed.
Comment 17
Within these bullet points on the main analyses, also, should be mentioned the possibility of using the non-parametric equivalents of independent t-tests and paired t-tests, in case the parametric assumptions are not met. These are only mentioned later on in the data analysis section.
Response 17
Thank you very much for this comment. This information has been added under the paragraphs “2.5. Statistical analysis” as follows: “For statistical inference, the non-parametric equivalents of independent t-tests and paired t-tests, were used in case the parametric assumptions were not met”.
Comment 18
RESULTS
Table 2 appears twice in my copy.
Response 18
We have removed the duplicated table. Thank you.
Comment 19
COMPARISON AT POST-INTERVENTION: Table 3 shows the results at post-intervention for the 2 groups clearly, thank you. Could you please indicate which were statistically significantly different?
Response 19
Table 3 is now Table 4 after changes in numbering. P-values have now been included in Table 4 to indicate the statistically significant differences at post-intervention comparisons between the 2 groups. Many thanks for this important comment.
Comment 20
COMPARISON AT PRE-INTERVENTION: Could you please add another table similar to Table 3, showing the results pre-intervention. From the answer to my question in the “response to authors”, I see that there were no differences in any of the outcome measures pre-intervention except for authoritarian and permissive parenting styles. This would be useful for readers to see.
Response 20
Thank you once again for this critical comment. We have included a new table (numbered Table 3) indicating the baseline (pre-intervention) comparisons between the intervention and the control group, which derived from the Independent t-test.
Comment 21
PHASE 2 or B PHASE: In the methods, “phase 2” referred to the last 3 sessions of the intervention. However, the results seem to use “phase 2” and “B phase” to refer to post-intervention results. Am I understanding this correctly? It would be clearer to use post-intervention throughout, when referring to the questionnaire results taken at the end of the study (rather than Phase 2 or B phase).
Response 21
This is corrected throughout the manuscript. Thank you.
Comment 22
TABLE 4: Please explain in text what “Authoritative Parenting Style plus new dim - Β phase Authoritative Parenting Style plus new dim” is.
Response 22
This was part of analytical work that shouldn’t be part of the manuscript. We have removed this information and apologize for the confusion.
Comment 23
PHASE 2 T-TEST RESULTS: In your response to reviewer (but not in the paper), you include 2 tables of t-tests, one at baseline and one at Phase 2 (which I take to be post-intervention). These are good, thank you. I’m puzzled though, because the second of those tables shows significant differences for all outcomes post-intervention, except permissive parenting style. But Table 3 in your paper seems to show bigger differences between the groups in authoritarian and permissive parenting styles than in any of the other variables. Just looking at the Table 3 results, I would have guessed that there might be significant differences between groups on authoritarian and permissive parenting styles, but certainly not on any of the other outcomes. So the t-test results don’t appear to match the descriptive data. Am I misunderstanding something here?
Response 23
Indeed, all the study dimensions demonstrate statistically significant differences at post-intervention level, except “permissive parenting style”. We have run the analysis again to confirm this information. Table 4 (Table 3 before the revisions) includes the correct information, which answers to this comment.
Comment 24
CLUSTER ANALYSIS: I still cannot understand what this analysis contributes to the paper. From the results so far, it appears to me that the intervention might have had an effect on authoritarian and permissive parenting styles (in Table 3), but not on any of the other outcomes (in Tables 3 and 4). However, the conclusion from the cluster analysis is that “the intervention program had a notable impact on these aspects of the participants' lives”. How could these 2 results both be true? They are inconsistent with one another.
Response 24
Thank you for this comment. We acknowledge that the cluster analysis, used in this study as supplementary to other statistical tests, may not be widely employed in these types of studies. However, we have already mentioned that our study found an increase in all dimensions at post- intervention level, but they were not found to be statistically significant. Our basic motivation for using cluster analysis, was thus to re-examine the post-intervention performance of the intervention group, using a different data classification and analysis, in order to be able to capture these changes.
To facilitate the readers’ understanding of this analysis, we have explained in detail the cluster analysis rationale under the paragraph “2.5 Statistical Analysis” and we have also included relevant information under Section “4. Discussion”.
Comment 25
Moreover, the abstract presents only this conclusion (not the results from the traditional analysis, showing little or no difference on most of the outcome measures). This conclusion doesn’t seem to be sufficiently justified from the data.
Response 25
We have re-written the abstract to better present the methods-results and conclusions of the study.
Comment 26
DISCUSSION
The Discussion is based on the conclusion drawn from the cluster analysis, and I’m not convinced that this is the correct conclusion from the data that has been presented.
The limitations need to place more weight on the level of attrition. It not only limited the analyses. It probably also resulted in a more biased sample, because participants do not drop out randomly. Two thirds of the control group dropped out. This is an extremely high rate, especially for a group who didn’t have to anything except complete 2 questionnaires at the end of 6 weeks. It deserves serious consideration.
Response 26
Thank you very much. This is the final and most crucial comment. We have re-written certain parts of the “Discussion section” to indicate that the study resulted in improvements in terms of parenting and quality of life dimensions but without these being statistically significant. We further improved the “study limitations” to refer to attrition and the biases that may have been introduced. Finally we referred to the need for further research.
Reviewer 2 Report
Comments and Suggestions for Authors
The authors did a great revised, but be carefully that it still It still has many formatting errors.
Author Response
Thank you very much